# A Multibreed Genome-Wide Association Study for Cattle Leukocyte Telomere Length

**DOI:** 10.3390/genes14081596

**Published:** 2023-08-07

**Authors:** Alexander V. Igoshin, Nikolay S. Yudin, Grigorii A. Romashov, Denis M. Larkin

**Affiliations:** 1The Federal Research Center Institute of Cytology and Genetics, The Siberian Branch of the Russian Academy of Sciences (ICG SB RAS), 630090 Novosibirsk, Russia; 2Royal Veterinary College, University of London, London NW1 0TU, UK

**Keywords:** leukocyte telomere length, cattle, genomewide association study, whole-genome resequencing

## Abstract

Telomeres are terminal DNA regions of chromosomes that prevent chromosomal fusion and degradation during cell division. In cattle, leukocyte telomere length (LTL) is associated with longevity, productive lifespan, and disease susceptibility. However, the genetic basis of LTL in this species is less studied than in humans. In this study, we utilized the whole-genome resequencing data of 239 animals from 17 cattle breeds for computational leukocyte telomere length estimation and subsequent genome-wide association study of LTL. As a result, we identified 42 significant SNPs, of which eight were found in seven genes (*EXOC6B*, *PTPRD*, *RPS6KC1*, *NSL1*, *AGBL1*, *ENSBTAG00000052188*, and *GPC1*) when using covariates for two major breed groups (Turano–Mongolian and European). Association analysis with covariates for breed effect detected 63 SNPs, including 13 in five genes (*EXOC6B*, *PTPRD*, *RPS6KC1*, *ENSBTAG00000040318*, and *NELL1*). The *PTPRD* gene, demonstrating the top signal in analysis with breed effect, was previously associated with leukocyte telomere length in cattle and likely is involved in the mechanism of alternative lengthening of telomeres. The single nucleotide variants found could be tested for marker-assisted selection to improve telomere-length-associated traits.

## 1. Introduction

The development of the genome-wide association study (GWAS) approach, in which the values of a phenotypic trait are compared with genome-wide variation in the studied population, has made it possible to improve the understanding of the genetic contribution to complex phenotypes [1,2]. Due to cost reduction, high-throughput whole-genome sequencing (WGS) is a popular alternative to array-based genotyping in GWASs [3]. Commercial SNP arrays mainly contain genetic variants, frequent in populations involved in their development [4]. Compared with array-based genotyping, whole-genome sequencing offers the advantage of analyzing rare and unique genetic variants. The rare variants in GWAS help solve the problem of complex traits’ “missing heritability” [5]. In addition, raw sequencing data can be used to evaluate quantitative characteristics of the genome, such as the average telomere length (TL) [6], copy number variation [7], mitochondrial or ribosomal DNA copy numbers [8,9], etc.

Telomeres are the DNA regions found at the ends of chromosomes. Together with specific nucleoprotein complexes, they safeguard the chromosomes from degradation during cell division [10]. Telomeric DNA consists of tandem repetitive DNA sequences (e.g., TTAGGG hexanucleotide in vertebrates) several thousand base pairs in length [11]. Since DNA polymerase does not fully replicate the 3′-ends of a DNA, telomeres are shortened by 50–200 nucleotides with each new cell division [12]. Two main antagonistic biochemical pathways regulate telomere length in normal cells. Firstly, a ribonucleoprotein complex called “telomerase” prevents telomere shortening through a reverse transcription reaction [13]. Secondly, the shelterin complex inhibits telomere elongation [14]. The 3′-end of the telomere, consisting of a guanine-rich single-stranded DNA region of 150–200 base pairs, is known to interact with the double-stranded region, forming the so-called T-loop. Since shelterin proteins stabilize the formation of the T-loop, repair proteins cannot recognize the terminal end of the chromosome. In addition, the T-loops inhibit telomerase activity by preventing access to the 3′ tail.

The issue with the end replication leads to the truncation of telomeric DNA during each cell division. As a result, telomere length shortens with age in humans and most other animal species [15,16], probably except naked mole rats (*Heterocephalus glaber*) [17]. Telomere shortening triggers the activation of DNA damage response, which involves cell aging and signaling pathways. Eventually, this results in cell cycle arrest, apoptosis, and progressive tissue atrophy [18,19]. Therefore, TL has been proposed as an individual’s life expectancy and health status predictor. Indeed, short TL correlates with reduced life expectancy in humans [20], mice [21], sheep [22], dogs [23], wild birds [24], and other species [25]. TL is associated with myocardial infarction, hypertension, Parkinson’s disease, cancer, diabetes, and other age-dependent pathological conditions in humans [26,27,28].

One of the goals of cattle selection is a productive lifespan [29]. However, selection progress is limited by complications in phenotype measurement and a low trait heritability [30]. Because dairy cows are usually culled when in poor health or fertility, their productive lifespan differs substantially from the longevity of humans and wild species [31]. TL, however, could be a proxy phenotype for productive lifespan and health measured at an early age, which would help to improve selection. Just like in most other animals, the telomeres of cattle tend to shorten as animals age [32,33]. It has been shown that adult Holstein cows with short telomeres are more likely to be culled in the subsequent year than cows with long telomeres [34]. In another study, productive life expectancy in Holsteins was positively correlated with TL at the age of one year [35]. Later, the same authors showed that the length of cattle’s productive life is associated with TL at birth [36]. Finally, they concluded that TL attrition was the best predictor of productive lifespan compared with an average TL or TL at the age of one year [37]. Apart from Holsteins, there is additional evidence for a link between TL and productive lifespan. According to Iannuzzi et al. [33], telomere length in the Agerolese cattle breed, showing a long productive lifespan, is significantly higher than in Holsteins, considering animals of the same age.

It is known that a multitude of age-independent pathological conditions related to immunity and fertility are associated with TL. TL at birth was associated with the risk of mastitis in Holstein cows [36]. Short TL was found to be over-represented among patients with unsuccessful pulmonary tuberculosis treatment outcomes [38] and patients with AIDS [39]. TL was also associated with several forms of female and male infertility in humans [40] and semen quality in bulls [41].

The TL trait is influenced by both genetic and environmental factors, making it a complex phenotype. It is known that TL differs significantly between *Caenorhabditis elegans* strains [42], inbred laboratory mouse strains [43], and dog [23] and cattle [33,44] breeds. TL heritability in Holstein–Friesian cattle ranges from 0.32 to 0.38 [35]. In another study, the monthly heritability of TL in Holsteins ranged from 0.36 to 0.47 [45]. A GWAS in Holstein–Friesian cattle identified 14 candidate genes for TL at birth and 9 at first lactation [36]. However, the genomic architecture of many quantitative traits in livestock differs substantially across various breeds [46,47]. Therefore, the GWAS of TL in non-Holstein breeds may help to gain a complete view of the genetic basis of TL variation in cattle.

Our study aimed to undertake seqGWAS among 17 cattle breeds, combining commercial and native populations with different genetic and environmental backgrounds to identify TL-associated genetic variants and confirm the effects of loci reported in previous studies.

Our recent work summarizing the results of 18 genome-wide association studies for telomere length in four species, including humans, showed that the overlap of candidate genes between species is virtually absent, except *pot-2* (*Caenorhabditis elegans*) and *POT1* (*Homo sapiens*) orthologs [48]. At the same time, multiple human studies show that the top associated genes (e.g., *TERC*, *TERT*, *STN1*, *POT1*, *DCAF4*, *RTEL1*, and *NAF1*) are consistently detected in samples from different ethnic groups [48]. At first sight, this observation could suggest that TL-control genes differ across animals but are the same within different populations (breeds) of the same species. To some extent, this assumption is reasonable. However, it should be remembered that the level of genetic variation between genes may differ and often is low for genes with high effect, thus affecting their detection by GWAS. Therefore, we hypothesize that relying on GWAS exclusively leads to underestimating the number of TL control genes shared by different species. As per our hypothesis, the study revealed that a few genes associated with telomeres were also related to telomere function in species other than cattle. In other species, however, this correlation has been identified through methods alternative to GWAS.

## 2. Materials and Methods

### 2.1. Samples and Sequencing

Whole-blood-extracted DNA genome resequencing data for 239 animals with known age and sex from 17 cattle breeds, of which 15 are reared in Russia, were used for variant calling and leukocyte telomere length estimation (Appendix A). Thirteen of these breeds were fully or mostly of European origin (Alatau, Bestuzhev, Charolais, Hereford, Kholmogory, Kostroma, Kazakh Whiteheaded, Black Pied, Red Steppe, Holstein, Tagil, Ukrainian Grey, and Yaroslavl). Four breeds had predominantly Turano–Mongolian genetic backgrounds (Buryat, Kalmyk, Wagyu, and Yakut) [49]. Resequencing data for 86 individuals were downloaded from the NCBI database (see Appendix A for SRA IDs), whereas 153 samples were resequenced (paired-end 2 × 150 bp) by Illumina’s Novaseq6000 technology at Novogene Co., Ltd. (Hong Kong, China) to ~50 Gbp each (~15× coverage). The latter samples’ DNA was extracted using cell lysation and the phenol–chloroform method. Before sequencing, DNA quality was assessed with agarose gel electrophoresis and by measuring 260/280 ratio with a BioTek Epoch spectrophotometer.

### 2.2. Variant Calling and Data Filtering

The samples were processed using the 1000 Bull Genomes Project guideline [50]. Briefly, raw sequencing reads were cleaned with Trimmomatic v.0.38 [51] and then mapped to the reference cattle genome (ARS-UCD1.2) using BWA-MEM v.0.7.17 [52]. Duplicate reads were marked with Picard v.2.18.2 (http://broadinstitute.github.io/picard/ (accessed on 4 January 2023)), after which we performed a base quality score recalibration (using the 1000 Bull Genome Project’s ARS1.2PlusY_BQSR_v3 dataset). BAM files obtained after this step were used for leukocyte telomere length estimation. A variant calling procedure was performed with GATK v.3.8-1-0-gf15c1c3ef [53].

The gVCF files obtained were merged using the GenotypeGVCFs subroutine of the GATK package. The resulting VCF files were filtered to have only biallelic single nucleotide polymorphisms, to which a “hard filtering” procedure was then applied (GATK’s filtering expression “QD < 2.0||FS > 60.000||MQ < 40.00||MQRankSum < −12.5||ReadPosRankSum < −8.0”). Low-quality genotypes (GQ < 12) were removed using VCFtools v. 0.1.16 [54], and then loci were filtered for minor allele frequency (--maf 0.025) and missing call rate (--geno 0.1) using PLINK v.1.9 software [55]. As filtering for possible paralogous sequence variants, we removed SNPs with excessive heterozygosity (>70%) and high D-statistics values (|D| > 4) calculated using *HDplot_python.py* script [56].

### 2.3. Power Estimate

To check whether the sample size of 239 individuals was sufficient to detect significant associations with the TL phenotype, we performed power analysis with QUANTO v.1.2.4 software [57], assuming continuous outcome and additive mode of inheritance. Type I error rate (two-sided) was set to 1.2 × 10^−7^, as obtained from the effective number of tests (0.05/412,450), which was in turn estimated by LD pruning method (PLINK: --indep-pairwise 5000 100 0.3) [58] applied to 13,923,772 SNPs left after filtering.

### 2.4. Leukocyte Telomere Length Estimation

Leukocyte telomere length was estimated from BAM files using TelSeq software version 0.0.2 [59]. TL calculated by TelSeq is the cumulative length of telomere repeats and includes interstitial telomere repeats. We modified parameters in the TelSeq source code to adapt it to the cattle genome, which includes changing the number of chromosomal ends, read length, and total GC content (bp). The parameters TELOMERE_ENDS, READ_LENGTH, and GENOME_LENGTH_AT_TEL_GC were set equal to 60, 150, and 321,325,950, respectively. We calculated the latter parameter by measuring the total length of 150 base pair windows in the cattle genome (ARS-UCD1.2), with a GC content between 48% and 52%.

### 2.5. Association Analysis and Variant Annotation

A genome-wide association analysis was performed with EMMAX software (INTEL compiled version). EMMAX exploits a mixed linear model accounting for population stratification and relatedness between individuals [60]. Initial telomere length estimates were normalized using the “*rankit*” method [61], implemented in the “*rcompanion*” R package [62]. The transformed values were then used as a phenotype under study. The association analysis was performed for the 13,923,772 SNPs left after the filtering steps. The kinship matrix (BN matrix) was computed based on a subset of 45,212 SNPs obtained by LD pruning (PLINK: --indep-pairwise 5000 100 0.1).

We conducted two types of GWAS, both including age and sex as covariates, but differently dealing with the multibreed composition of our dataset. In the first type of analysis, one additional covariate accounting for the influence of phylogenetic origin (European/Turano–Mongolian) on telomere length was added. For the second test, we included covariates for each breed’s effect using a “dummy coding” scheme [63] to account for between-breed phenotypic differences. To estimate the inflation of test statistics, λ_GC_ factor was calculated [64]. To control for the multiple testing error rate, the false discovery rate (FDR) was applied using the “*qvalue*” R function [65]. Q-values of 0.05 and 0.10 were the thresholds for significant and suggestive levels, respectively. Single nucleotide polymorphisms showing significant or suggestive association levels were annotated with SnpEff software [66]. Where needed, a search for gene homologs was carried out using the BlastP algorithm [67]. If a candidate gene was associated with telomere length in another study, which utilized a different bovine assembly, the corresponding variant coordinates were remapped to the ARS-UCD1.2 positions using the LiftOver tool [68].

To confirm that the effects of candidate SNPs are independent of age and breed, we regressed the initial and normalized LTL values on age and breed covariates, obtained model residuals, which were further utilized as the adjusted phenotypes, and then analyzed their association with doses of minor alleles using the “*lm*” R function.

## 3. Results

### 3.1. Variant Calling and Data Filtering

In total, 53,240,581 polymorphisms were detected with the GATK HaplotypeCaller. The consecutive removal of indels, multiallelic sites, and low-quality variants resulted in 46,734,220, 45,835,735, and 41,705,108 SNPs, respectively. The filtering for missing genotype rate, minor allele frequency, heterozygosity, and D-statistics removed 7,187,088, 20,225,052, 46,283, and 322,913 SNPs, which resulted in 13,923,772 variants further used in association testing. Their genome distribution was similar to that in other mammals [69,70], with distal chromosome regions having higher SNP density (Appendix A).

### 3.2. Power Estimate

According to a previous GWAS in Holstein cattle, top SNPs with additive effect explained 10% and 11% of total phenotypic variance in telomere length [36]. Our statistical power estimation analysis showed that the sample size of 239 animals has the power of 0.39 and 0.49 to detect loci with such effects, respectively. This statistical power level is typically regarded as moderate and, in actual use, is enough to uncover associations between genotype and phenotype [71,72,73,74].

### 3.3. Leukocyte Telomere Length Estimation

The estimated leukocyte telomere lengths ranged from 4.2 Kbp to 84.3 Kbp (Appendix A). The highest values were observed in the Kazakh Whiteheaded (mean length: 27.2 Kbp, mean age: 2.7 years), Hereford (22.5 Kbp, two years), and Buryat (22.7 Kbp, 3.3 years) breeds. The cattle breeds with the lowest telomere lengths were the Wagyu, measuring 6.9 Kbp and 2.3 years, and the Ukrainian Grey, measuring 6.1 Kbp and 8.2 years. Overall, the estimated telomere length was negatively correlated with the animal’s age (Pearson’s r = −0.21, *p*-value < 0.01; Spearman’s ρ = −0.305, *p*-value < 1 × 10^−5^).

### 3.4. Association Analysis and Variant Annotation

The QQ plots and λ_GC_ factor values suggest the absence of statistical inflation (Appendix A). Genome-wide association testing with breed covariates (Figure 1, Appendix A) resulted in 63 statistically significant (q-value < 0.05) SNPs found on 21 chromosomes. The top association signal (BTA8:36516223, *p*-value = 2.78 × 10^−11^, q-value = 2.5 × 10^−4^) was found in the *PTPRD* gene. Of the 63 SNPs detected, 3 were downstream gene variants, 3 were upstream gene variants, 44 were found in intergenic regions, 12 were intronic variants, and 1 was a missense variant. The intron and missense SNPs were found in five protein-coding genes (intron: *PTPRD*, *EXOC6B*, *RPS6KC1*, and *NELL1* and missense: *ENSBTAG00000040318*) (Table 1). At the suggestive significance level, there were 56 additional SNPs: downstream (1), intergenic (25), intron (29), and upstream (1) variants.

The analysis with breed group covariate identified 42 SNPs (Figure 1, Appendix A) located on 15 chromosomes, with the strongest signal (BTA29:44612473, *p*-value = 1.89 × 10^−15^, q-value = 2.63 × 10^−8^) found in the *CTSF*-*CCDC87* intergenic region. Overall, the SNPs found in this type of analysis were downstream gene variants (2), intergenic region variants (31), intron variants (8), and upstream gene variants (1). The intron variants were found in seven protein-coding genes (*EXOC6B*, *PTPRD*, *NSL1*, *RPS6KC1*, *AGBL1*, *ENSBTAG00000052188,* and *GPC1*) (Table 1). At the suggestive significance level, 52 additional SNPs were detected: downstream (3), upstream (2), intergenic (22), intron (24), and missense (1) variants.

Minor allele frequencies (MAF) of significant SNPs were in the range of 0.025–0.228 and 0.025–0.186 for the “breed effect” and “group effect” designs, respectively. However, most loci (84% in the former analysis and 95% in the latter) had MAF < 0.05. The number of breeds in which an associated SNP was polymorphic ranged from 2 to 17 (Appendix A). In general, when the minor allele frequency of an SNP is higher, it tends to segregate in a larger number of breeds.

We conducted a point-based association analysis on the top gene *PTPRD*, previously linked to leukocyte telomere length in cattle [36]. The analysis was performed using adjusted phenotypes to test SNPs. We selected two SNPs that were found to be significant in both types of our GWAS analysis, namely BTA8:36496551 and BTA8:36516223. Our analysis shows that the minor alleles of these SNPs are associated with reduced LTL independently of animals’ age and breed (Figure 2). This is observed both for initial and normalized LTL values. Other SNPs from the *PTPRD* gene should have similar effects, as they are in strong linkage disequilibrium (r^2^ = 0.85–1.0) with BTA8:36496551 and BTA8:36516223.

## 4. Discussion

In this study, we examine the genetic structure of telomere length in various taurine cattle breeds by analyzing their whole-genome resequencing data. This improves the likelihood of identifying causative variants compared with SNP arrays [75]. Including multiple breeds in the design could lead to more accurate QTL mapping, as the linkage disequilibrium patterns vary between individual breeds [76]. Also, our experiment was designed to point to alleles with effects across multiple breeds [77] and avoid breed- or family-specific effects [78]. Our dataset consists of animals from various populations and environments. Therefore, the variants identified may play a role in different environments where taurine cattle can be found.

Human studies have shown that the length of telomeres at birth is a reliable indicator of a person’s telomere length throughout their life, from childhood to adulthood [79]. As has been shown for Holstein cattle, however, there is only a weak correlation (Pearson’s r = 0.30) between leukocyte telomere length values at birth and at first lactation [36], which suggests that different genetic mechanisms could be involved at different ages, at least in this breed. Since our study involves animals of varying ages, we anticipate that some associated variants will have a lasting impact throughout their lifespan. Our SNPs could therefore be tested as markers of TL in breeding strategies.

The findings of this study agree with previously published data. According to a microarray-based GWAS in Holstein–Friesian cattle, the *PTPRD* gene was the closest to the most significant SNP associated with telomere length at first lactation [36]. In our study, SNPs from the *PTPRD* were detected in both experimental designs. Additionally, within our breed design, this gene exhibits the most significant statistical signal. However, according to remapping results, the distance between the variant from the abovementioned study and ours is 1.13 Mb. This falls within the range of distances between causal polymorphism and observed association signals [80]. This is even less surprising considering an extensive level of linkage disequilibrium in Holstein–Friesian cattle resulting from a high level of inbreeding and intense artificial selection in this breed [81]. The PTPRD is a tumor suppressor often inactivated and mutated in tumors [82,83]. The exact way this protein could affect telomere length is unknown but is likely related to the alternative lengthening of telomeres (ALT), representing a way to maintain telomeres unrelated to telomerase activity. Thus, according to Hartlieb et al. [84], deletions in *PTPRD* are enriched in ALT-positive tumors. The GWAS in humans confirms the participation of genes from the ALT pathway in telomere length variation, though their contribution is relatively minor [85].

All detected SNPs from the *PTPRD* gene segregate in at least four breeds: Charolais, Hereford, Kazakh Whiteheaded, and Red Steppe. In Holsteins, however, none of the associated SNPs were polymorphic. This may be connected to the limited number of animals engaged (15) and/or genetic differences between Holstein populations used in the study by Ilska-Warner et al. [36] and ours. As was confirmed by the additional analysis of adjusted phenotypes, the top *PTPRD* gene’s SNPs have breed- and age-independent modes of action. Therefore, we could propose them for further validation and subsequent marker-assisted selection to improve telomere-length-associated traits in the breeds where this haplotype segregates.

Another gene with evidence of possible involvement in TL maintenance is *GPC1*, which has been found in our “group effect” GWAS. This gene encodes for glypican-1 protein belonging to a family of heparan sulfate proteoglycans. Liu et al. [86] demonstrated that genes differentially expressed between *GPC1* high- and low-expressing colon adenocarcinoma tumors are enriched by signaling pathways related to telomere maintenance.

The *NSL1* gene may have a potential role in controlling telomere length. It encodes for the NSL1 component of the MIS12 kinetochore complex. As shown by Zhang et al. [87], instability of the MIS12 complex accelerates the senescence of human mesenchymal progenitor cells (hMPCs). Moreover, *MIS12*-knockout hMPCs have an approximately four-fold decrease in telomere length.

In addition to genes with significant SNPs, there were also potential gene candidates that showed association signals nearby and others that had SNPs at a suggestive significance level. Thus, we detected a significant SNP upstream of *ANKRD17*, a telomere-associated gene responsible for longevity in naked mole rats [88]. Another candidate is *CPSF4*, with an intronic variant detected at the suggestive level. According to Chen et al. [89], CPSF4 is a promoter-regulating protein of human telomerase reverse transcriptase (hTERT) capable of enhancing telomerase activity in lung cancer cells and normal lung cells.

The only missense variant found (BTA18:47718019, Lys1887Met) was in *ENSBTAG00000040318*—the uncharacterized gene. The *ENSBTAG00000040318* gene shares 93.5% amino acid sequence identity with the *WDR87* gene of *Bubalus bubalis* (92% cover). This gene encodes for testis development protein NYD-SP11 (https://www.genecards.org/cgi-bin/carddisp.pl?gene=WDR87 (accessed on 22 July 2023)). We found no link between *WDR87* and telomeres. However, according to a comparative phylogenetic study by Muntane and colleagues [90], WDR87 is associated with lifespan in primates. The telomere shortening rate is closely linked to lifespan, implying that *WDR87* may play a role in telomere biology [16].

Overall, the relation to telomere biology for genes identified in this study was previously shown in cattle (*PTPRD*), humans (*GPC1*, *NSL1*, *CPSF4*), and naked mole rats (*ANKRD17*). In agreement with our hypothesis, candidate genes with relation to telomeres in noncattle species had their possible involvement in telomere biology demonstrated by non-GWAS techniques. This stresses the importance of using distinct methodologies for understanding telomere-related mechanisms and pathways shared by animal species.

Considering the relatively small number of genes identified in this study, it is hard to resolve the predominant mechanisms and pathways underlying the genomic architecture of bovine leukocyte telomere length variation. Based on the information collected here and in previous studies, the *PTPRD* gene may significantly affect telomere length variation in cattle. This could indicate that alternative lengthening of telomeres plays a substantial role in this species. This is also supported by the fact that genes known to be directly related to telomerase have not been detected at the level of significance in our study. On the other hand, according to a study in humans, top association signals harbored by core components of proteins regulating the assembly and activity of the telomerase mostly come from common variants with small effects [85]. Therefore, it is possible that due to a smaller sample size leading to moderate statistical power, SNPs in such genes have not been detected in our study.

## 5. Conclusions

Our study utilizing whole-genome resequencing of diverse cattle samples reveals several genomic loci associated with bovine leukocyte telomere length. The top results coupled with the previous finding suggest that the *PTPRD* gene might be one of the major contributors to leucocyte TL variation in cattle. The detected variants could be further tested in marker-assisted selection to improve telomere-length-associated traits such as longevity, productive lifespan, stress resistance, and disease susceptibility in cattle.

## Figures and Tables

**Figure 1 genes-14-01596-f001:**
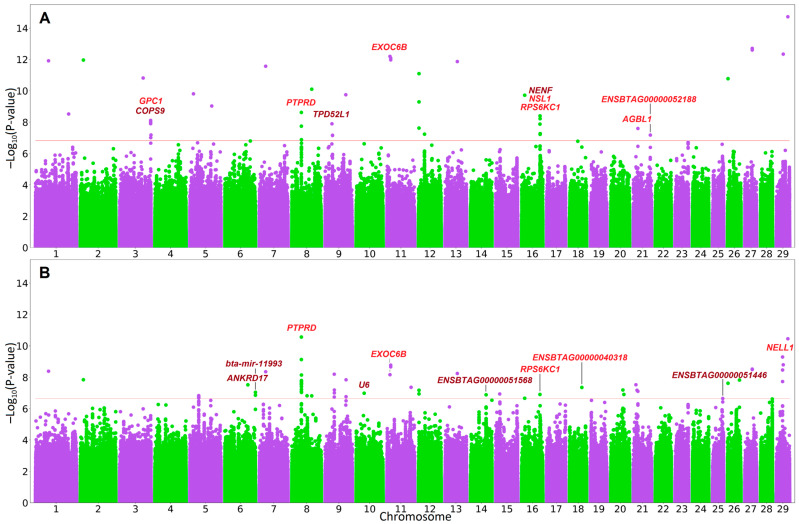
Manhattan plots for genome-wide association analysis with covariates for a group of breeds (**A**) and breed effect (**B**). Solid horizontal lines indicate a significant threshold (q-value = 0.05). In red are genes harboring associated SNPs. In brown are genes with significant upstream or downstream variants.

**Figure 2 genes-14-01596-f002:**
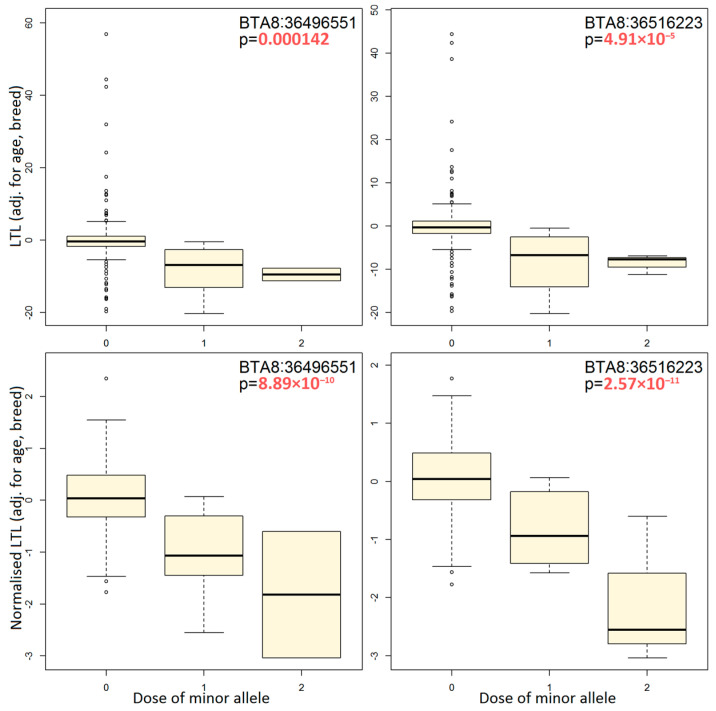
Boxplots illustrating the effect of two top SNPs from the *PTPRD* gene using the raw and normalized telomere lengths adjusted for age and breed.

**Table 1 genes-14-01596-t001:** Genic SNPs significantly associated with LTL in at least one type of analysis.

Position (BTA)	Ref/Alt Allele	Gene	Variant Type	q-Value ^1^	q-Value ^2^
3:119873308	A */G	*GPC1*	intron variant	3.31 × 10^−2^	NA
8:36400561	C */T	*PTPRD*	intron variant	NA	1.45 × 10^−2^
8:36433838	T */G	*PTPRD*	intron variant	NA	2.79 × 10^−2^
8:36441863	A */C	*PTPRD*	intron variant	NA	2.79 × 10^−2^
8:36496551	G */T	*PTPRD*	intron variant	8.15 × 10^−3^	2.57 × 10^−3^
8:36516223	G */A	*PTPRD*	intron variant	1.59 × 10^−3^	2.5 × 10^−4^
8:36554362	G */A	*PTPRD*	intron variant	NA	2.5 × 10^−2^
8:36583129	A */G	*PTPRD*	intron variant	NA	3.03 × 10^−2^
11:12087073	T/G *	*EXOC6B*	intron variant	1.77 × 10^−6^	6.21 × 10^−3^
16:70386226	T/C *	*RPS6KC1*	intron variant	6.22 × 10^−3^	3.59 × 10^−2^
16:70795459	G */T	*NSL1*	intron variant	3.48 × 10^−3^	8.89 × 10^−2^
18:47718019	T */A	*ENSBTAG00000040318*	missense variant	7.42 × 10^−2^	1.93 × 10^−2^
21:17460307	G/A *	*AGBL1*	intron variant	1.07 × 10^−2^	8.51 × 10^−2^
21:65476950	C/T *	*ENSBTAG00000052188*	intron variant	2.44 × 10^−2^	NA
29:24118907	T */C	*ENSBTAG00000048576 (NELL1)*	intron variant	NA	2.33 × 10^−3^
29:24142137	A */G	*ENSBTAG00000048576 (NELL1)*	intron variant	NA	4.79 × 10^−3^
29:24168343	A */G	*ENSBTAG00000048576 (NELL1)*	intron variant	NA	1.17 × 10^−2^

*—Allele contributing to higher LTL. ^1^—Design with covariate for groups of breeds. ^2^—Design with covariates for individual breeds.

## Data Availability

The whole-genome sequence’s raw data are available from the corresponding author on a reasonable request.

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
