# Peer review of "A Multibreed Genome-Wide Association Study for Cattle Leukocyte Telomere Length"

_genes, 2023, doi:10.3390/genes14081596_

Round 1

Reviewer 1 Report

I have 2 major remarks for this paper:

1) the authors should provide a clear hypothesis. At the moment, I can not find a clearly stated hypothesis.

2) the sample size. There is no clear sample size calculation provided. Can the authors comment on the sample size they have used in the study?

Author Response

We are grateful to Reviewer 1 for a positive review of our manuscript. Our responses to all comments are below:

Comment: the authors should provide a clear hypothesis. At the moment, I can not find a clearly stated hypothesis.

Response: We added the hypothesis that interspecies comparisons of candidate TL genes based on GWAS only lead to an underestimation of the number of shared genes between species (lines 111-125).

Comment: the sample size. There is no clear sample size calculation provided. Can the authors comment on the sample size they have used in the study?

Response: We added a power analysis for our sample size using Quanto software. It shows that our sample of 239 animals has moderate power (~0.4-0.5) to detect loci with strong effects (10-11% of variance explained) (lines 155-160, 217-222).

Reviewer 2 Report

Title: A Multi-Breed Genome-Wide Association Study for Cattle Leukocyte Telomere Length 

Main comment:

Your study aimed to undertake seqGWAS among 90 cattle breeds, combining commercial and native populations with different genetic and environmental backgrounds, to identify TL-associated genetic variants and confirm the effects of loci reported in previous studies. However, it is not clear if your study successfully achieved its goals. The connection between the PTPRD gene, TL, age, and cattle breed is not established. Several points remain unresolved, and the experimental design appears unclear. I suggest a major revision.

Line 70: There is a study on TL in cattle breeds that demonstrates the correlation between breed and longer telomeres compared to Holstein. Additionally, this study reports TL in older cattle (13 years) for the first time (https://doi.org/10.1111/age.13227).

Line 96: Can you provide more information about the matrix used for the analysis (blood, milk; fresh or old samples; storage conditions)?

Line 98: In Table S1, you reported TL estimations (Kb). Did you use an absolute method for these estimations (Column G)? Are they reproducible, yielding the same results in other laboratories?

Line 171: You classify cattle according to their breed in Table S1. I attempted to reclassify the S1 table based on cattle age, and there are significant discrepancies in the data. There are only 16 cattle with an age greater than 10 years, and among them, the TL estimations vary considerably (4.98 to 13.3). Can you provide a better explanation for these results? Telomeres are typically negatively correlated with age, as you and other recent articles have reported.

Line 214: I disagree with this statement. I suggest including more than one reference to support it. The reference you provided pertains to a single cattle breed, and it should be noted that TL is correlated with both breed and the matrix used for calculation (https://doi.org/10.1111/age.13227; https://doi.org/10.3390/d2091118).

Line 264: The PTPRD gene might be a major contributor to leucocyte TL variation in cattle...

Is the PTPRD gene also correlated with cattle breed? Is it also correlated with TL in relation to age?

Author Response

We would like to thank Reviewer 2 for a thoughtful review of our manuscript. Our point-by-point responses to all specific comments are below:

Comment: Line 70: There is a study on TL in cattle breeds that demonstrates the correlation between breed and longer telomeres compared to Holstein. Additionally, this study reports TL in older cattle (13 years) for the first time (https://doi.org/10.1111/age.13227).

Response: Thank you for pointing to this paper. We added the corresponding information to the text (lines 77-81).

Comment: Line 96: Can you provide more information about the matrix used for the analysis (blood, milk; fresh or old samples; storage conditions)?

Response: All DNA samples were isolated from blood. We noted this in the text (lines 119, 128-131). The samples we sequenced were fresh and checked for quality by gel electrophoresis and absorbance measurements at 260/280 before the sequencing lab QC. The quality of the rest of the samples, which we downloaded from the NCBI, was not specified. We filtered the data both at the SNP level and at the level of animal genotypes. Therefore, potential problems related to SNP quality should be eliminated.

Comment: Line 98: In Table S1, you reported TL estimations (Kb). Did you use an absolute method for these estimations (Column G)? Are they reproducible, yielding the same results in other laboratories?

Response: Yes, Telseq is an absolute method, which gives telomere length estimates in kilobases. These estimates correlate with qPCR (r=0.65) and Southern blot (r=0.6-0.71) measurements in different labs (Ding et al. 2014; Zhang et al. 2022). As far as we know, Telseq is a deterministic method which uses a simple formula for TL calculation (Ding et al. 2014), therefore its results should be the same across different runs.

References:

Zhang D, Newton CA, Wang B, Povysil G, Noth I, Martinez FJ, Raghu G, Goldstein D, Garcia CK. Utility of whole genome sequencing in assessing risk and clinically relevant outcomes for pulmonary fibrosis. Eur Respir J. 2022 Dec 22;60(6):2200577.

Ding Z, Mangino M, Aviv A, Spector T, Durbin R; UK10K Consortium. Estimating telomere length from whole genome sequence data. Nucleic Acids Res. 2014 May;42(9):e75.

Comment: Line 171: You classify cattle according to their breed in Table S1. I attempted to reclassify the S1 table based on cattle age, and there are significant discrepancies in the data. There are only 16 cattle with an age greater than 10 years, and among them, the TL estimations vary considerably (4.98 to 13.3). Can you provide a better explanation for these results? Telomeres are typically negatively correlated with age, as you and other recent articles have reported.

Response: We have 17 individuals with age >10 years. Indeed, there is no negative correlation with age in this group. However, these 17 animals are represented by different breeds. If you look separately at these breeds, you will find out that the corresponding animals with age >10 years have shorter telomeres compared with animals <10 years with the only exception for the Bestuzhev breed samples, but there are only two samples from this breed for this age range:

Yakut (>10 years) = 9.86 Kb on average (6 animals), Yakut (<10 years) = 10.95 Kb on average (31 animals)

Bestuzhev (>10 years) = 9.62 (3), Bestuzhev (<10 years) = 8.54 (2 animals)

Kalmyk (>10 years) = 6.62 (3), Kalmyk (<10 years) = 8.42 (27 animals)

Tagil (>10 years) = 7.32 (2), Tagil (<10 years) =7.50 (10 animals)

Ukrainian Grey (>10 years) = 5.32 (2), Ukrainian Grey (<10 years) = 6.31 (8 animals)

Alatau (>10 years) = 5.55 (1), Alatau (<10 years) = 7.87 (2 animals)

Comment: Line 214: I disagree with this statement. I suggest including more than one reference to support it. The reference you provided pertains to a single cattle breed, and it should be noted that TL is correlated with both breed and the matrix used for calculation (https://doi.org/10.1111/age.13227; https://doi.org/10.3390/d2091118).

Response: We have not found any other studies analyzing the correlation between leukocyte TL in cattle at different ages. Therefore, we have noted in the text that these data are limited to the Holstein breed (lines 299, 302). Also, we noted that telomere length differs between cattle breeds (line 90). As we used only blood DNA-derived data, we did not mention TL correlation with the matrix.

Comment: Line 264: The PTPRD gene might be a major contributor to leucocyte TL variation in cattle... Is the PTPRD gene also correlated with cattle breed? Is it also correlated with TL in relation to age?

Response: The SNPs of the PTPRD gene are polymorphic in the Hereford, Kazakh Whitehead, Kalmyk, Red Steppe, and Charolais breeds. We have added a column containing information about breeds in which the SNP is polymorphic for all SNPs (see Table S2). The breed factor was considered in one of two types of GWAS analysis. Animals’ age was added as an additional predictor in both designs (group, breed). Therefore, alleles of SNPs from the PTPRD gene are correlated with telomere length independently of age.

You can look at the SNP chr8:36516223, the top in the PTPRD gene. Considering the age ≤1 year, the association between allele dosage and rank-transformed TL is significant: p-value = 0.015 in a linear model (“lm” R function), mean TL in minor allele carriers = 9.61 Kb, in major allele homozygotes = 15.28 Kb.

In the group of 4 to 8 years, in which the chr8:36516223 is polymorphic, p-value = 0.00231. Mean TL in minor allele carriers = 6.03 Kb, in major allele homozygotes = 10.52 Kb.

So, a minor allele of the SNP chr8:36516223 decreases TL independently of age. Similar observations can be made for other SNPs in the PTPRD gene. We added information about SNP genotypes in animals to supplements (Table S3).

Also, we conducted additional analysis to confirm that the effects of most important candidate SNPs are independent of age and breed (lines 199-202, 274-282).

Round 2

Reviewer 2 Report

After carefully considering my review, the authors responded in a remarkably adept and satisfactory manner, addressing all the points raised. Their thorough and thoughtful response demonstrated their commitment to addressing concerns and enhancing the quality of their work.